# Hierarchical Nanoflowers of MgFe_2_O_4_, Bentonite and B-,P- Co-Doped Graphene Oxide as Adsorbent and Photocatalyst: Optimization of Parameters by Box–Behnken Methodology

**DOI:** 10.3390/ijms23179678

**Published:** 2022-08-26

**Authors:** Manpreet Kaur Ubhi, Manpreet Kaur, Dhanwinder Singh, Mohammed Javed, Aderbal C. Oliveira, Vijayendra Kumar Garg, Virender K. Sharma

**Affiliations:** 1Department of Chemistry, Punjab Agricultural University, Ludhiana 141001, Punjab, India; 2Department of Soil Science, Punjab Agricultural University, Ludhiana 141001, Punjab, India; 3Department of Mathematics, Statistic and Physics, Punjab Agricultural University, Ludhiana 141001, Punjab, India; 4Institute of Physics, University of Brasilia, Brasilia 70000-000, Brazil; 5Program for Environment and Sustainability, Department of Environmental and Occupational Health, School of Public Health, Texas A&M University (TAMU), College Station, TX 77843-1266, USA

**Keywords:** hierarchical nanoflower, nanocomposite, boron and phosphorus doped GO, magnesium ferrite-bentonite, adsorbent, photocatalyst, Box–Behnken design

## Abstract

In the present study, nanocomposites having hierarchical nanoflowers (HNFs) -like morphology were synthesized by ultra-sonication approach. HNFs were ternary composite of MgFe_2_O_4_ and bentonite with boron-, phosphorous- co-doped graphene oxide (BPGO). The HNFs were fully characterized using different analytical tools *viz*. X-ray photoelectron spectroscopy, scanning electron microscopy, energy dispersion spectroscopy, transmission electron microscopy, X-ray diffraction, vibrating sample magnetometry and Mössbauer analysis. Transmission electron micrographs showed that chiffon-like BPGO nanosheets were wrapped on the MgFe_2_O_4_-bentonite surface, resulting in a porous flower-like morphology. The red-shift in XPS binding energies of HNFs as compared to MgFe_2_O_4_-bentoniteand BPGO revealed the presence of strong interactions between the two materials. Box–Behnken statistical methodology was employed to optimize adsorptive and photocatalytic parameters using Pb(II) and malathion as model pollutants, respectively. HNFs exhibited excellent adsorption ability for Pb(II) ions, with the Langmuir adsorption capacity of 654 mg g^−1^ at optimized pH 6.0 and 96% photocatalytic degradation of malathion at pH 9.0 as compared to MgFe_2_O_4_-bentonite and BPGO. Results obtained in this study clearly indicate that HNFs are promising nanocomposite for the removal of inorganic and organic contaminants from the aqueous solutions.

## 1. Introduction

There is a surging research interest in the fabrication of nanocomposites of ferrites with promising adsorptive and photocatalytic properties. Different morphologies of nanocomposites *viz*. flower-like [1,2], hollow-yolk shell [3], core-shell [4] and nanorods [5] are reported. These nanostructures have some unique features such as fast charge transfer, the reduced recombination rate of charge carriers and high surface area which allows their utilization in adsorption and photocatalysis [6]. For enhancing the activity of these nanocomposites, the right choice of the components is an important aspect.

In recent years, graphene-based materials have been extensively studied for the environmental applications because of their large surface area and tunable optical properties [7]. Graphene oxide (GO), a functionalized form of graphene, has adsorptive and catalytic inertness because of its low chemical potential and restricted surface functionalities [8]. GO is reported to have a wide bandgap (3.5 eV) and can act as a photocatalyst only under UV light illumination [8]. This feature limits its practical applications for visible light assisted photocatalysis. Visible radiation constitutes 45% of solar light and its use in photocatalysis makes it a green technology for the remediation of wastewater [9,10]. Co-doping of GO with heteroatoms can improve its properties for application as adsorbent and photocatalyst. Co-doping of GO with boron, nitrogen, sulphur and phosphorous has been reported [11,12,13]. These co-doped materials have been extensively used in electrocatalysis due to asymmetric charge density induced by the difference between the electronegativity of heteroatoms and carbon [14,15,16].

The boron and nitrogen co-doped reduced GO has been reported as an effective catalyst towards the electrochemical degradation of paracetamol, due to the improvement of conductivity and creation of new surface defects [17]. Nitrogen and sulfur co-doped industrial graphene showed high peroxymonosulfate activation for catalyzing the methyl paraben oxidation [15]. Similarly, the nitrogen-sulfur co-doped reduced GO nanohybrids displayed high adsorption capacity for cationic, anionic and neutral dyes [18]. Improved catalytic and adsorptive features of co-doped GO can be due to large specific surface area, pore size and pore volume as compared to mono-doped and pristine GO. Although boron, nitrogen, and sulfur co-doped GO/graphene have been studied [15,17], B- and P- co-doped GO and its composites have not yet been explored as adsorbents and photocatalysts for the removal of pollutants from the aqueous media.

In this work, boron and phosphorous heteroatoms were chosen for co-doping of GO lattice because they can induce a synergistic coupling effect due to the co-existence of electron-deficient boron atoms and electron-rich phosphorous atoms. They also exhibit excellent optical properties along with mesoporosity and large surface area. Different bonding configurations of boron and phosphorous atoms offer huge adsorptive sites for the uptake of pollutants. The recovery of co-doped GO after adsorption and photocatalysis is difficult due to its non-magnetic nature. This problem can be overcome by making its nanocomposite with ferrites. They are mixed metal oxides of iron with promising adsorptive, photocatalytic and magnetic properties. Magnesium ferrite (MgFe_2_O_4_)-bentonite nanocomposite has been reported as an effective adsorbent and photocatalyst in our previous studies [19,20,21]. In the present work, nanoflowers of MgFe_2_O_4_-bentonite nanocomposite with boron and phosphorous co-doped GO (BPGO) were synthesized, which led to enhancement in the overall adsorptive and photocatalytic activity as compared to both the components. The structural, magnetic, and optical features of the synthesized nanoflowers were analyzed using different analytical techniques. BPGO nanosheets wrapped on the MgFe_2_O_4_-bentonite surface produced a hierarchical nanohybrid with flower-like morphology. The adsorptive and photocatalytic potential of the synthesized nanoflowers was evaluated using Pb(II) and malathion as model pollutants. The concentration of Pb(II) and malathion should be less than 0.015 ppm and 100 ppm in the wastewater, as per WHO guidelines [22]. Presence of Pb(II) in water above permissible limit can cause anaemia, hypertension, renal impairment, immunotoxicity and neurological disorders. Malathion (Diethyl 2-[(dimethoxy phosphorothioyl) sulfanyl] butanedioate) has been broadly used in agriculture to control weeds and can cause renal failure, myocardial depression, central nervous system disorders, lung edema, and eye irritation [23,24,25,26].

Box–Behnken Design (BBD) was utilized as a statistics tool for optimizing the adsorption and photocatalytic parameters and for exploring the simultaneous effects of the independent variables on the dependent variables such as Pb(II) removal and malathion degradation. Further, kinetics of the adsorption and degradation process was studied in detail. The adsorption mechanism was predicted on the basis of XPS studies, thermodynamic parameters, adsorption isotherm and kinetic modelling. A plausible photodegradation mechanism was also proposed on the basis of GC-MS analysis and the free radical quenching studies. Our work presents an efficient method to synthesize hierarchical recyclable adsorbent and photocatalyst with promising potential application in water decontamination.

## 2. Results and Discussion

### 2.1. Characterization

Table 1 lists all the information about the synthesized samples, including sample codes and their description.

#### 2.1.1. Structural Characterization

XPS spectrum of MGF-B (Figure 1a) confirmed the existence of both MgFe_2_O_4_ and bentonite as revealed by the existence of Mg, O, Si, Na, Fe and Al. The binding energy levels at 48 eV and 1304 eV attributes to Mgՙ2p’ and Mgՙ1s’, respectively indicated the divalent oxidation state of Mg (Figure 1b,c). The XPS spectrum of Feՙ3p’ can be split into two sub-peaks at binding energies of 55 eV and 53 eV showing the presence of Fe(III) form in MGF-B (Figure 1b). Two major peaks associated with the Fe_1/2_ and Fe_3/2_ electronic levels, as well as satellite peaks associated with shake-up, were observed in the Feՙ2p’ XPS spectra (Figure 1d) [27]. The deconvolution of Feՙ2p’ spectrum exhibited six sub-peaks of Fe in 3/2 and 1/2 state in Oh, Td and satellite peaks. High resolution Oՙ1s’ spectrum (Figure 1e) depicted the existence of OH attached to metal ion (530 eV) and O−Fe(Mg) (528 eV). High-resolution XPS spectrum for the Alՙ2p’ (Figure 1f), Naՙ1s’ (Figure 2a) and Siՙ2p’ (Figure 2b) core level at 72 eV, 1071 eV and 103 eV binding energy, respectively depicts the presence of all minerals of bentonite in the magnesium ferrite-bentonite nanocomposite. The full XPS survey scan of BPGO depicted the presence of Cՙ1s’, Oՙ1s’, Bՙ1s’ and Pՙ2p’ elements. High resolution spectrum of Oՙ1s’ energy levels showed peaks at 529 eV and 531 eV which were assigned to C=O and C-O/P-O energy positions (Figure 1e). The fitted peaks for C ՙ1s’ located at 283 eV to 286 eV were ascribed to C-C, C-P and C-O moieties (Figure 2c). The Bՙ1s’ spectrum can be spilt into two peaks for in-plane BC_3_ type bonding (189 eV) and borinic ester (C_2_BO)/boronic acid (CBO_2_) moieties (191 eV), indicating that the B heteroatoms exist in three distinct chemical environments (Figure 2d). The deconvoluted spectra of Pՙ2p’ showed two peaks at binding energies of 133 and 134 eV corresponding to the P-C and P-O species, respectively (Figure 2e). These findings showed presence of both B and P containing moieties in GO lattice.

The HNFs’ XPS full scan spectrum confirmed the existence of all elements of BPGO and MGF-B, indicating its fabrication. However, all the characteristic peak positions were slightly changed due to different coordination environments, which in turn caused a change in the observed binding energy. Strong interactions between MGF−B and BPGO were indicated by the red shift in binding energies across all elements when compared totwopristine materials. During the catalytic processes, these interactions favour fast migration and charge carrier separation caused by light. Table 2 lists the atomic concentration of each element.

Mössbauer spectrum of MGF-B (Figure 2f) displayed two sextets and one quadruple doublet. One outer sextet is assigned to Fe^3+^ ion in octahedral (O_h_) sites and other in tetrahedral (T_d_) sites [28]. The quadrupole splitting value of both the sextets is very small, indicating the symmetrical environment around Fe^3+^ ions. Additionally, a quadruple doublet was foundwhich was ascribed to the super paramagnetic relaxation of small particles with a particle diameter less than 15 nm. These small particles are magnetically isolated and do not participate in the long-range magnetic ordering [29,30].

There are more Fe^3+^ ions in the Td sites, according to the relative area of the sextets (Table 3). On insertion of BPGO into MGF-B, the area of absorption attributing to the superparamagnetic part (doublet) decreased which may be due to the formation of nanoflowers, as they have large size (8.6 nm) as compared to MGF-B (4.0 nm). Size of NPs has a significant impact on the Mössbauer spectrum. The presence of broad sextets and doublet occurs when the relaxation rate and Larmor precession frequency are of the same order of magnitude in the case of larger NPs [31]. HNFs displayed a similar Mössbauer spectrum, confirming the presence of MgFe_2_O_4_.

FT-IR spectrum of HNFs (Figure 3) confirmed the existence of different B- and P- containing functional groups, M-O bond and Si-O framework vibrations due to BPGO, MgFe_2_O_4_ and bentonite, respectively. The details are given in the Appendix A.

The XRD pattern of MGF-B (Figure 3b) displayed all the peaks of MgFe_2_O_4_ (2theta = 30.1°, 35.4°, 43.1°, 53.5°, 57.0° and 62.6° indexed to hkl planes of (220), (311), (400), (422), (511) and (440), respectively). XRD peak of bentonite at 2theta = 26.6° (003) did not appear in the XRD pattern of MGF-B due to a very low amount of bentonite clay in it or exfoliation of clay. Our previous work showed that with increasing bentonite clay in MGF-B sample, the bentonite peak was clearly observed [20,32].

In GO, the XRD diffraction peaks at 2theta values of 10.9° and 42.7° can be indexed to (001) and (101) hkl planes of graphitic carbon, respectively (Appendix A) [33]. After co-doping (Appendix A), the fingerprint diffraction peak of GO at 2θ = 10.9°disappeared. Low intensity and broad diffraction peaks at 2theta = 25.0° and 43.6° referring to (002) and (101) crystallographic planes, respectively, were observed. The diffraction peak at 2theta = 25.0° was due to highly disordered graphitic carbon structure [34]. The peak observed at 2theta = 43.6° was most probably due to the defects in the GO lattice.

In the HNFs (Figure 3b), the XRD peaks correspond to MGF-B and BPGO are retained, while their peak intensities are changed due to interactions between them. According to Bragg’s equation, the d-spacing of most intense peak of MGF-B, BPGO and HNFs at 2theta ≈ 35.4°, 25.0° and 35.5° was 0.25, 0.34 and 0.24 nm, respectively. The decrease in d-spacing of HNFs can be ascribed to the creation of more defective sites, which results in an increased number of smaller graphitic domains. Based on the Debye Scherrer’s formula [35] and the peak at full width half-maximum of MGF-B (311) and HNFs (311) crystal plane, the average crystallite size was about 4.0 and 8.6 nm, respectively (Table 2).

#### 2.1.2. Morphological Studies

A TEM micrograph of MGF-B (Figure 4a) revealed the existence of spherical and agglomerated MgFe_2_O_4_ NPs over the bentonite matrix. The agglomeration was due to the nanosized crystallites and magnetic character of the ferrite NPs. The average grain size was 30–35 nm. TEM image of BPGO showed folded/scrolled chiffon like nanosheets and twisted edges (Figure 4b). After integrating MGF-B with BPGO nanosheets, the as-synthesized nanocomposite depicted nanoflower like microspheres with a 3D structure, produced from the interlayer Van der Waals forces and high surface energy of MGF-B (Figure 4c). In the presence of MGF-B, the nanostructures of BPGO gradually transform from the chiffon like nanosheet to nanoflower-like shapes. Chiffon-like BPGO nanosheets with wrinkled surface were wrapped on the MGF-B surface, producing a hierarchical nanohybrid with porous structure. Along with nanoflowers morphology, the nanocomposite having MGF-B particles on the BPGO surface was also observed (inset of Figure 4i).

A SEM micrograph of MGF-B (Figure 4d) confirmed that MgFe_2_O_4_ NPs were incorporated into the bentonite layers. Similar micrographs have been observed upon the insertion of polypropylene into the layers of bentonite clay [19]. The SEM micrographs of BPGO depict a translucent and rippled silk waves with numerous crinkles possibly due to dopants and oxygeneous groups. These wrinkled nanosheets adjust themselves physically to adapt to MGF-B (Figure 4e). SEM micrographs of HNFs displayed sponge-like morphology with structure like nanoflowers (Figure 4f).

The atomic and weight percentage of all the elements present in the synthesized materials was depicted via energy-dispersive spectroscopy (EDS) (Figure 4g–i). They confirmed the presence of different elements in MGF-B, BPGO and HNFs. The peak for B was not observed in the EDS spectra as lighter elements does not possess Bremsstrahlung scattering [36]. The elemental composition of each element found in the synthesized materials is listed in Table 2.

The Brunauer-Emmett-Teller (BET) surface area of HNFs calculated using the nitrogen adsorption-desorption technique was 187.56 m^2^g^−1^ which was higher than magnesium ferrite-bentonite (87.10 m^2^g^−1^) and BPGO (162.8 m^2^g^−1^). This may be ascribed to the inhibited agglomeration of pristine MGF-B in the presence of BPGO. The BPGO and HNFs followed type IV isotherm with H3 hysteresis loops indicating the existence of mesoporous particles (diameters 2–50 nm) according to IUPAC classification, whereas MGF-B indicates type IV isotherm with a H4 typical hysteresis curve, confirming the existence of mesoporous particles with internal slit-like pores [37] (Figure 5a). The pore size distribution curves of MGF-B, BPGO and HNFs (Figure 5a inset) exhibited sharp narrow peaks in the range of 3.11 nm to 3.89 nm (Table 2). The pore volume of MGF-B, BPGO and HNFs was in the range of 0.07–0.29 cm^3^g^−1^, respectively. Thus, the enlarged surface area and pore diameter of HNFs as compared to MGF-B can be ascribed to the presence of BPGO. It is a favourable feature for the adsorptive and photocatalytic process.

#### 2.1.3. Thermal, Magnetic and Optical Studies

TGA-DTG-DTA analysis of HNFs, BPGO and MGF-B displayed stability of these materials up to 500 °C which signifies their practical applications (Figure 5b). Details are discussed in Appendix A.

Magnetic field dependent hysteresis curves of the MGF-B and HNFs are shown in Figure 6a. The values of saturation magnetization (M_s_), coercivity (H_c_) and remanence (M_r_) are given in Table 2. This clearly indicated that a significant decline in M_s_ value of HNFs (6.26 emu g^−1^) was observed after its fabrication with BPGO. Thus, fabrication of BPGO nanosheets on the surface of MGF-B altered the magnetic features as it decreased the M_s_ value. Due to their ferrimagnetic nature, these materials showed a remarkable affinity for the external magnet. The decrease in H_c_ value for the HNFs (70.65 Gauss) from MGF-B (138.00 Gauss) confirmed that less magnetic field was required to demagnetize it. This decrease showed enhancement in its soft magnetic feature.

UV-VIS diffuse reflectance (UV-Vis DRS) spectroscopy was used to determine the band gap of synthesized materials (Figure 6b). MGF-B, BPGO, and HNFs’ optical band edges (E_g_) were calculated to be 2.26, 2.40, and 2.10 eV, respectively. The chemical interactions between MGF-B and BPGO, which were supported by a decline in binding energy in XPS analysis, may be the reason of the HNFs’ band edges narrowing. The absorbance of all the materials in the visible light region is of great significance in the area of visible light photocatalysis.

The photoluminescence emission spectra in the wavelength range of 200–900 nm was studied for MGF-B, BPGO and HNFs (Figure 6c). Emission peaks were observed at 430 nm for MGF-B and 720 nm for BPGO and 410, 720 nm for HNFs. Emission peaks at 410 and 430 nm were attributed to 3d^5^→3d^4^ transitions in Fe^3+^ and may be caused by radiative defects or surface imperfections [38]. After its fabrication with BPGO, five-fold decrease in luminous intensity was observed, confirming higher quenching efficiency of HNFs over pristine materials. The findings also suggested that the transfer of electrons from the conduction band of the photo-excited MGF-B to BPGO sheets might slow down the photoinduced charge carrier recombination over pristine materials, resulting in the enhanced photo-catalytic degradation of organic pollutants.

### 2.2. Adsorption Experiments

HNFs were observed to be better adsorbent for Pb(II) ions than BPGO and MGF-B. Detailed explanation is given in Appendix A. HNFs combined the features of both MGF-B and BPGO, which enhanced their adsorption potential. The major factor for the enhancement in adsorption potential could be the increase in the surface area and porosity as depicted by BET analysis. It was further supported by DLS studies, which showed that the particle size of HNFs (130 nm) in solution was smaller than MGF-B (171 nm). The HNFs possessed appreciable M_s_ value, which led to their easy separation and reusability. Box–Behnken statistical analysis was applied to optimize the adsorption parameters for Pb(II) ions on HNFs.

#### 2.2.1. Statistical Analysis of Adsorption Studies

The randomised design matrix of the Box–Behnken statistical model with 4 factors, 3 central points, and 27 runs as well as the removal efficiency of the Pb(II) ions, are listed in Table 4 and their ANOVA results shown in Table 5. The present study’s independent variable *p*-values were less than 0.05, proving that the experimental data may sufficiently characterize the proposed model derived using the Box–Behnken response surface methodology. The predicted and experimental statistical results were found to agree, and the correlation coefficient (R^2^, 0.99) and adjusted R^2^ (0.98) values were also quite high. The model’s validity is further supported by the 0.86 *p*-value of lack-of-fit, which was found to be non-significant. This value was higher than the lowest limit of fit as recommended to be (0.05). The model’s calculated value for adequate precision was 37.50, which supported a greater S/N ratio. Any process design that has a signal (response) to noise (deviation) ratio of >four is preferable, and the model can be moved around in the design space. According to Table 5, A, B, C, D, AB, AC, AD and BD are significant terms for removing Pb(II) ions. Thus, the reduced expression of the quadratic regression model is presented as below:Pb(II) removal (%) = +90.0 + 16.0 A + 4.79 B + 0.04 C + 5.0 D + 0.12 AB − 1.0 AC + 0.23 AD −1.29 BD + 1.56 A^2^ + 1.01 C^2^

This expression demonstrates the empirical relationship between significant variables and Pb(II) removal (response). The plot (Appendix A) between normal probability and externally studentized residuals follows a normal distribution where the points lie on a straight line. The reliability of the assumptions and the independence of the residuals are indicated by this plot. The relationship between the actual and predicted values (Appendix A) shows that the experimental findings for this study arewell-accepted. All of these findings point to the proposed model’s strong correlation and suitability for predicting the Pb(II) adsorption process utilizing the HNFs as an adsorbent.

#### 2.2.2. Effect of pH, Adsorbent Dosage and Contact Time

The solution pH, adsorbent dosage and contact time played an vital role in adsorption of Pb(II) ions. In addition to affecting the charge on the adsorbent surface, the pH of the solution also affects the equilibrium between adsorption and desorption and the surface charge of Pb(II) ions. The results plotted in Figure 6d–f demonstrate that the percentage removal of Pb(II) ions increased progressively from 61% to 95% with rise in the solution pH from 2.0 to 6.0. At a lower solution pH, the surface adsorptive sites got protonated and became positively charged thus inducing an electrostatic repulsion for Pb(II) ions, whereas under alkaline conditions, the surface became more negative, providing the electrostatic attraction for Pb(II) ions. The pH_ZPC_ studies (Appendix A) revealed surface charge on HNFs surface became zero at pH 4.3. Above pH 4.3, Pb(II) ions were strongly attracted towards the negatively charged surface resulting in maximum uptake of Pb(II) ions. Speciation of Pb(II) in aqueous solution using visual miniteq software showed that free ionic Pb(II) is the most predominated form (80–90%) up to pH 6.0, whereas Pb(OH)_2_ (90–100%) dominates after this pH (Appendix A). Under alkaline conditions, Pb(II) started to form insoluble Pb(OH)_2_ and the solubility of the metal hydroxides decreased at higher pH. These results were further supported by ξ-potential studies. The ξ-potential of HNFs at pH 6.0 was negative (Appendix A). The negative ξ-potential of HNFs was responsible for the electrostatic attraction between positively charged Pb(II) ions and negatively charge HNFs surface.

The adsorption of Pb(II) ions enhanced on increasing the nanoadsorbent dose up to 0.4 g L^−1^ due to more available surface area and binding sites for the removal of Pb(II) ions. After that, no increase was observed (Figure 6d and Figure 7a) because of attainment of adsorption-desorption equilibrium. Adsorption was monitored as a function of contact time also. It followed three stages i.e., fast, slow and equilibrium stage (Figure 6e). The fast stage lasted for first 5 min, as the solute molecules freely attached to the available active sites. The slow stage persisted for up to 10 min. The equilibrium stage was attained after 30 min where insignificant difference in the percentage removal was observed.

#### 2.2.3. Kinetic and Adsorption Isotherm Modelling

Comparative evaluation of linear and non-linear form of kinetic modelling was employed for determining the best fitting model. Linear pseudo second-order kinetic model (Figure 7b) provided higher ‘R^2^’ value of 0.99 and lower statistical errors than its non-linear form (R^2^ = 0.64) (Appendix A) suggested that the adsorption data followed the linear form of modelling. The better fitting of pseudo second-order kinetic model than pseudo-first order (Appendix A),intra-particle diffusion (Appendix A) and Elovich (Appendix A) specified the interaction of Pb(II) ions with two binding sites of the HNFs. Detailed explanation of kinetics results and parameters are given in Appendix A, respectively.

Adsorption isotherms, both linear and non-linear, were used to analyse the data from the adsorption equilibrium. The statistical functions shown in Appendix A were used to compare the two regressions. The linear Langmuir equation’s goodness-of-fit over non-linear form was defined by its maximum “R^2^” (0.99) and lowest error values (Figure 7c). The separation factor ‘R_L_’ was in the range of ‘0.0–1.0’ depicting the adsorption of Pb(II) ions was favourable in nature. The q_max_ value for Pb(II) ions was 745.4 mg g^−1^ and it was compared with other reported adsorbents (Appendix A). Other adsorption isotherms are explained in detail in Appendix A. The decreasing order of goodness-of-fit of employed isotherms was as follows: Langmuir > Freundlich > Temkin > Dubinin-Radushkevich. Thus, Pb(II) ions interacted uniformly with the active sites of HNFs in the monolayered physical adsorption shown by the adsorption isotherm modelling.

Similar outcomes for the Pb(II) ions adsorption employing CaFe_2_O_4_-NGO nanocomposite were observed by Kaur et al. [22]. According to Buergisser et al. [39], natural adsorbents usually follownon-linear models. However, batch tests for the Cr (VI) adsorption over Rhizopus sp. conducted by Espinoza-Sánchez et al. [40] showed that linear regression provided a better fit. Yazdani et al. [41] showed the goodness-of-fit of the non-linear method for batch As (V) adsorption utilizing nano-TiO_2_/feldspar-embedded chitosan. However, in the current investigations, nanocomposite used linear regression and contained both natural and synthetic components. Thus, model fitting is dependent upon the nature of the adsorbent. Therefore, using both linear and non-linear modelling is essential for any adsorbent.

#### 2.2.4. Thermodynamic and Regeneration Studies

The adsorption capacity of Pb(II) on HNFs increased with an increase in reaction temperature (Figure 6f and Figure 7a) and could reach 94% at 50 °C. However, the adsorption capacity decreased above 50 °C. This may be due to the increase in average kinetic energy of ions with temperature leading to increase in adsorption as more number of ions interacted with the mesoporous surface of the nanostructure. After that, desorption dominates, leading to decline in their adsorption potential. The values of ΔG° was negative, indicating that the adsorption of Pb(II) on the HNFs was spontaneous (Appendix A). The absolute values of ΔG° increased as the temperature rose, indicating an increase in spontaneity. The endothermic nature of Pb(II) was illustrated by a positive ΔH° value, with an optimal temperature of 50 °C. The entropy change values (ΔS°) were positive, indicating that as the reaction temperature increased, the degree of freedom of the adsorption system also increased (Appendix A).

The excellent recycling performance of adsorbent determines its application in practical environment. As seen in Appendix A, after five adsorption-desorption cycles, the 89% removal efficiency was retained which confirmed reusability of HNFs. Moreover, the magnetic character facilitated their separation from the solution using an external magnetic field.

#### 2.2.5. Effect of Co-Existing Cations and Anions

The adsorption capacity of the HNFs was tested in the presence of Pb(II), Cd(II), Zn(II) and Ni(II) ions (Appendix A). The observed trend for the removal was Pb(II) > Cd(II) > Zn(II) > Ni(II) and can be described by distribution and selectivity studies. The distribution coefficient and selectivity factor listed in Appendix A indicated that the adsorption of Pb(II) ions was preferred in the existence of other metal ions. The adsorption behaviour was also explained by ionic radii of the metal ions (Appendix A). The larger the ionic size of ions, the lower is their hydrated radii hence have a greater affinity to get adsorbed. The adsorption capacity of HNFs for Pb(II), Cd(II), Zn(II) and Ni(II) ions decreased in the quaternary system than single system due to competition between metal ions for the adsorption sites available on HNFs surface.

The effect of presence of anions *viz*. Cl^−^, NO_3_^−^, SO_4_^2−^, CO_3_^2−^ and PO_4_^2−^ was also tested on the Pb(II) adsorption. The increase in anion concentration from 0.01 to 10.0 mM showed a negative effect on metal ions adsorption (Appendix A). However, this negative impact is maximum in the presence of PO_4_^2−^ ions followed by CO_3_^2−^, SO_4_^2−^, NO_3_^−^ and Cl^−^ ions. This may be explained by the interaction of ligands with heavy metals to create ion pairs (Kaur et al., 2019). The availability of Pb(II) ions for adsorption decreases with increasing ion-pair stability. Here, PO_4_^2−^ and Pb(II) ions form the most stable ion pair, which reduces the amount of Pb(II) that is adsorbed from the Pb_3_(PO_4_)_2_ solution.

### 2.3. Photocatalytic Degradation Studies

The photocatalytic potential of MGF-B, BPGO and HNFs using malathion as the model pollutant was evaluated. An insignificant increase in malathion degradation was observed from pH 1.0 to 3.0 (Appendix A), which indicated its excellent stability in an acidic medium as reported previously [42]. At basic pH, Organophosphate pesticides are more prone to oxidation and hydrolysis due to the presence of hydroxide ion (OH^−^). Bavcon et al. [43], Zhao and Hwang [44] reported decomposition of malathion in the similar pattern. Thus, the optimum pH for degradation of malathion is 9.0.

The photocatalytic ability of HNFs was higher than BPGO and MGF-B. This improvement has been supposed to be due to (a) huge specific surface area, lowering in aggregation, easy charge transfer and layered structure of BPGO. The co-doped GO nanosheets act as a supporting substrate for the deposition of MGF-B, which enhanced the uptake capacity of the GO based photocatalyst.

The reduced aggregation resulted in higher specific surface area of HNFs (187.56 m^2^g^−1^) as compared to BPGO (162.81 m^2^g^−1^) and MGF-B (87.1 m^2^g^−1^) which provides more uptake sites for the pollutant molecules. (b) small size of HNFs (47 nm) as observed by DLS size distributions as compared to BPGO (60 nm) and MGF-B (77 nm) favoured higher adsorption, (c) Another factor was reduced recombination rate of electron-hole pair.

A control test was performed under visible light exposure in the pH range of 1.0 to 9.0. Appendix A indicated poor self-degradation ability of malathion due to their stability in the absence of photocatalyst. The degradation was 38% under visible light illuminations which was significantly enhanced up to 99% in the presence of HNFs. Control experiment in the absence of light demonstrated less than 25% removal of malathion when treated with HNFs. The findings imply that photocatalytic degradation employing synthesized materials as a photocatalyst was mainly responsible for the malathion removal.

On increasing the photocatalyst dose from 0.04 to 0.2 g L^−1^, the percentage of photodegradation was enhanced. Further increase in photocatalyst dosage displayed a decrease in degradation of pollutants. This may be ascribed to (1) the presence of huge specific surface area and active sites. The aggregation of HNFs, which results in the low penetration of visible light, may be the reason of the decline in photodegradation effectiveness at higher dosages.

#### 2.3.1. Effect of Contact Time and Kinetic Studies

The photocatalytic decomposition of malathion with time was deduced by using UV-VIS absorption spectra (Figure 7f). The absorbance of base peak at 234 nm for malathion attenuated with the time, confirming its degradation. The photocatalytic degradation of malathion followed the Langmuir Hinshelwood first-order model. The value of apparent rate constant (k) calculated from ln(C_t_/C_o_) vs. time was found to be 2.36 × 10^−2^ min^−1^ (Figure 8a,b). Similar results were reported by Fakhri and Bagheri (2020) during the degradation of malathion and tetracycline using UiO@metal oxide/GO as a photocatalyst.

#### 2.3.2. Statistical Analysis of Photodegradation Studies

A randomized design matrix and ANOVA for Box–Behnken regression model of malathion degradation is given in Table 6 and Table 7, respectively. The terms F-value (79.73) and *p*-value (0.0001) are displayed in the ANOVA. Values less than 0.05 confirm that the given factors are significant at the 5% level of confidence. F test = 5.36 (*p* = 0.75) for “Lack of Fit” revealed that the model was adequately fitted and there was no lack of fit. The model’s calculated value for adequate precision was 25.55, which supported a greater S/N ratio. The predicted quadratic regression model’s R^2^ (coefficient of determination) value was 99.7. As a result, this model may be used to more accurately predict the response at any level of the selected factors. The reduced expression of the quadratic regression model is presented below:Malathion degradation (%) = +77.0 + 33.5 A − 4.13 B + 4.63 C − 3.25 AB + 0.25 AC − 21.25 A^2^

Appendix A displays the uniform distribution of data along a straight line and their strong correlation. The residuals of removal percentage are often plotted on a straight line in the normal probability plot (Appendix A). This demonstrates the validity of the normal distribution and proves to the model’s applicability for accurate data fitting. Highest malathion degradation (%) was achieved at the maximum level (+1) of pH (9.0), minimum level (−1) of photocatalyst dose and median level (0) of contact time (Figure 7d,e).

Photocatalytic activity using malathion as model pollutant at pH ranging from 3.0 to 9.0, with variation in contact time = 30 to 120 min and variation in photocatalyst dose from 0.2 to 0.8 g L^−1^. Figure 7d shows that as the photocatalytic activity increased with increasing pH and contact time, thestrong photocatalytic activity occurs at basic pH values and at a certain photocatalyst dose (Figure 7e). This could be a result of the catalyst particles aggregating at larger doses, which results in the low visible light penetration.

#### 2.3.3. Quenching Studies

In order to evaluate the reactive species involved in the decomposition of organic pollutants, four quenchers *viz*. ascorbic acid, sodium azide, ethylene triamine tetraacetic acid and methanol were used as superoxide (O_2_^−^), singlet oxygen (O_2_*), holes (h^+^) and hydroxide (OH) quenchers, respectively (Figure 8c). The kinetic studies in the presence of quenchers indicated the photocatalytic degradation was significantly inhibited in presence of sodium azide and ascorbic acid. The addition of ascorbic acid produced the greatest inhibition, indicating that O_2_^•−^ radicals are essential to the photodegradation process [45]. Here, it can be observedthat the reaction between holes and superoxide radicals produces singlet oxygen species. Contrary to these findings, the addition of EDTA and methanol slightly inhibited the degradation activity, confirming that h+ and ^•^OH radicals also played a minor role in the photocatalytic degradation of malathion.

#### 2.3.4. Analysis of Degraded Products

Degradation products formed at the end of the irradiation process were identified by GC-MS analysis (Appendix A). Oxidative desulfuration of malathion (*m*/*z* = 330) resulted in the formation of malaoxon (*m*/*z* = 314), which was further hydrolyzed into malaoxon monoacid (*m*/*z* = 287) and diethyl 2-mercaptosuccinate (*m*/*z* = 207) *via.* a carboxyl ester hydrolysis and a competing elimination reaction, respectively. When malathion was hydrolyzed, charged intermediates were created by the nucleophilic addition of ^−^OH to the phosphorus atom. Malaoxon, which has a P=O moiety, is more likely to hydrolyze than malathion, which has a P=S moiety [20]. This enhances nucleophilic substitution by the ^−^OH, which in turn encourages hydrolysis [46]. Diethyl 2-mercaptosuccinate was easily breakdown into two fragments *viz.* ethyl acetate (*m*/*z* = 88) and ethylmercaptoacetate (*m*/*z* = 120). The fragments with *m*/*z* = 94, PO_4_^3−^; *m*/*z* = 44, CO_2_; *m*/*z* = 18, H_2_O; *m*/*z* = 32, CH_3_OH and *m*/*z* = 60, CH_3_COOH resulted from oxidative degradation of diethyl 2-mercaptosuccinate, ethylmercaptoacetate and ethyl acetate. Thus, degradation product analysis reveals degradation into smaller fragment molecules along with CO_2,_ and H_2_O. A similar degradation pathway was reported for MgFe_2_O_4_-bentonite nanocomposite [46]. However, HNFs have the advantage of higher photocatalytic potential.

#### 2.3.5. Reusability Studies

One of the key factors determining a synthesizedphotocatalyst’s effectiveness and practical usability is its stability. The reusability test of HNFs was run for seven consecutive cycles in order to assess the photocatalyst stability. After each cycle, the used photocatalyst was centrifuged from reaction vessel through centrifugation and washed with distilled water and used again forthe next cycle. A minimal loss in the percentage degradation was observed. The photocatalytic efficacy of HNFs decreases by ~5%, after five consecutive cycles (Appendix A) indicating excellent stability of the HNFs photocatalyst.

### 2.4. Adsorption and Photodegradation Mechanism

HNFs as nanoadsorbent (Figure 9a) has the combined features of magnesium ferrite-bentonite (Path-A) and BPGO (Path-B) as:(1)The adsorption of Pb(II) ions on magnesium ferrite-bentonite part is mainly driven by Van der Waals forces, ion-exchange andelectrostatic interactions due to its negative charge.(2)BPGO having intrinsically negatively charged surface due to phosphoanhydride, boronic, borinic acid, hydroxyl and carbonyl moieties which provides electrostatic interactions to Pb(II) ions.(3)The agglomeration of magnesium ferrite-bentonite and restacking of BPGO layers was diminished on interactions of BPGO layers with magnesium ferrite-bentonite, which reduced the required effective dosage of nanoadsorbent for adsorptive removal of Pb(II) ions than pristine materials.(4)HNFs possessed appreciable M_s_ value which led to its easy separation and reusability, which affected the cost of the adsorption process.

XPS and FT-IR spectra of HNFs were assessed after adsorption of Pb(II) ions. Full scan XPS spectrum of Pb(II) adsorbed HNFs (Figure 8d) depicted the presence of all elements of magnesium ferrite-bentonite and BPGO, along with Pb(II) peaks. After the adsorption of Pb(II) ions, the binding energy values of all the elements showed a minor positive shift, indicating a change in the local bonding environment. The XPS peaks at binding energies of 136 eV and 141 eV correspond to Pb4f_7/2_ and Pb4f_5/2_, respectively (Figure 8e). The interaction of Pb(II) ions with the HNFs consumed the complex boronic, P-O and oxy functional moieties by the release of water molecules from its coordinated sphere.

In the FT-IR spectrum of Pb(II) adsorbed HNFs (Figure 8f), the bands affirmed to B- bonding, P- bonding and M-O vibrations were observed with reduced intensity and at lower wavenumbers which might be attributed to the electrostatic interactions of Pb(II) ions with these moieties present on the HNFs. All the bands except C=C and C-P shifted towards lower wavenumber due to complex formation of Pb(II) ions with the lone pair present on the oxygen atom of the adsorbed hydroxyl groups from the aqueous phase via electron sharing. XPS and FT-IR studies provided the strong evidence for the adsorption of metal ions on the surface of HNFs.

A probable photocatalytic mechanism for the degradation of malathion has been proposed (Figure 9b). Upon irradiation of light on the photocatalyst surface, the electrons present in the valance band are excited to the conduction band of magnesium ferrite-bentonite (Equation (1)). These photoinduced electrons can be readily accepted by BPGO nanosheets (as a matrix) to suppress the charge carrier recombination, which in turn improves the photodegradation ability (Equation (2)). Here, co-doped nanosheets played two vital roles for the improvement of the photocatalytic ability of the synthesized nanostructure. Briefly, the planar π-π framework of co-doped nanosheets makes it a good electron accepting material. The electrons present in the conduction band of magnesium ferrite-bentonite can readily move to co-doped nanosheets through percolation mechanism. This helps in decreasing recombination rate of charge carriers and thus more electrons are accessible for the generation of reactive species, boosting photocatalytic degradation. A good conducting nature of co-doped nanosheets allowed fast charge carrier transport, which helped in their effective carriers separation during the photocatalytic reaction. During this transfer of electrons, molecular oxygen reduces to form superoxide radical anion (O_2_^•−^) (Equation (3)). Singlet oxygen species were generated by the reaction of holes with O_2_^•−^ radicals (Equation (4)). The holes situated in the valence band oxidize hydroxyl ions (OH^−^) or water molecules (H_2_O) to produce hydroxyl radicals (HO^•^) (Equation (5)).The resulting reactive oxygen species degraded organic pollutants into the degradation products i.e., CO_2_ and H_2_O (Equation (6)). The degradation mechanism can be expressed as:(1)$$\mathrm{Magnesium}\;\mathrm{ferrite}-\mathrm{bentonite}+\mathrm{hv}\to \mathrm{Magnesium}\;\mathrm{ferrite}-\mathrm{bentonite}\;(e^{-}+h^{+})$$
(2)$$\mathrm{BPGO}+\mathrm{Magnesium}\;\mathrm{ferrite}-\mathrm{bentonite}\;(e^{-})\to \mathrm{BPGO}\;(e^{-})$$
(3)$$\mathrm{BPGO}\;(e^{-})+O_{2}\to {O_{2}}^{-}+\mathrm{BPGO}$$
(4)$${O_{2}}^{-}+\mathrm{Magnesium}\;\mathrm{ferrite}-\mathrm{bentonite}\;(h^{+})\to {O_{2}}^{*}$$
(5)$${\mathrm{HO}}^{-}+\mathrm{Magnesium}\;\mathrm{ferrite}-\mathrm{bentonite}\;(h^{+})\to {\mathrm{OH}}^{.}$$
(6)ROS+Organic pollutants→Degraded products

Hence, BPGO because of its better transporting and electron accepting nature combined with a united effect of magnesium ferrite-bentonite make the nanocomposite photocatalysts visibly active.

The magnesium ferrite-bentonite bound by the hexagonal BPGO hexagonal arrays, offered strong conductive network and electrical channels for magnesium ferrite-bentonite, thereby ensuring the structural integrity and enabling the effective transfer of electrons and holes during photocatalysis. The fabrication of BPGO with magnesium ferrite-bentonite offered more specific surface area (187.56 m^2^g^−1^) for the hierarchical nanohybrid ensuring the effective interaction space between organic contaminants and active sites.

## 3. Methods and Materials

### 3.1. Reagents

All the chemicals (analytical reagent grade), such as graphite powder, orthoboric acid, orthophosphoric acid, sulphuric acid, hydrogen chloride, malathion, sodium nitrate, potassium permanganate, hydrogen peroxide, metal nitrates, bentonite, ethylene diammine tetraacetic acid, sodium azide, ascorbic acid, methanol and sodium hydroxide were procured from SD fine Pvt. Ltd. Deionized water was used to prepare all the solutions.

### 3.2. Synthesis and Characterization

#### 3.2.1. Synthesis of Boron and Phosphorous Co-Doped Graphene Oxide (BPGO)

BPGO was synthesized using orthoboric acid and orthophosphoric acid as the source for B and P, respectively. 50 mL of aqueous dispersion of GO (20 mg mL^−1^) was vigorously stirred with orthoboric acid (5.7 g) and orthophosphoric acid (1.60 mL) for 2 h at 25 °C. The solution was evaporated to dryness. Remnant solid was annealed in the muffle furnace at 300 °C for 3 h to obtain BPGO. The aqueous suspension of BPGO was ultra-sonicated for 6 h, centrifuged and then washed with hydrogen chloride solution (5%) to eliminate unreacted reactants. Finally, it was washed with distilled water several times to get neutral pH followed by ethanol and dried at 80 °C overnight.

#### 3.2.2. Synthesis of Hierarchical Nanoflowers (HNFs)

HNFs were synthesized in two steps. In the first step, binary composite of MgFe_2_O_4_ and bentonite (MGF-B) was synthesized by sol-gel method, reported in our previous publication [20,21] which was further transformed into ternary hierarchical nanoflowers in the second step by adding BPGO to this binary composite. MGF-B and BPGO aqueous solutions were separately prepared by ultrasonication at room temperature. Finally, the ternary HNFs were collected by centrifugation and dried. The HNFs were synthesized in different *w*:*w* ratios (MGF-B:BPGO; 0.5:1, 1:1 and 2:1). The prepared samples were characterized by various physicochemical techniques discussed in Appendix A.

### 3.3. Pollutants Removal Evaluation

#### 3.3.1. Adsorption Experiments

To perform batch mode adsorption experiments, stock solution of Pb(II) ions of 100 mg L^−1^ was prepared using Pb(NO_3_)_2_·4H_2_O. The first experiment was performed to determine the best adsorbent from MGF-B, BPGO and HNFs using 5 mg L^−1^ concentration of Pb(II) ions (100 mL) and 0.4 g L^−1^ of adsorbents at 25 °C. Then, using the best nanoadsorbent, next adsorption experiments were performed. To obtain the maximum adsorption efficiency, the batch mode adsorption experimental conditions, which included pH (2.0–10.0), contact time (2–120 min), adsorbent dosage (0.2–0.8 g L^−1^), and temperature (10–55 °C), were tuned. For pH adjustment, HCl or NaOH solutions were used. The adsorbate-adsorbent solutions were centrifuged and the concentration of Pb(II) in the centrifugates was evaluated using the ICAP-OES technique after shaking (240 rpm) on an orbital shaker for a fixedperiod of time (2 h). Limits of quantification (LOQ) and limit of detection (LOD) were also assessed for Pb(II) ions as explained in Appendix A. The adsorption efficiency (AE) and adsorption capacity (q_t_) of HNFs was calculated as:$$\mathrm{Adsorption}\;\mathrm{effciency}\;(\mathrm{AE})\;=\frac{C_{i}-C_{t}}{C_{i}}\times 100$$
$$\mathrm{Adsorption}\;\mathrm{capacity}\;(q_{t})\;=\frac{C_{i}-C_{t}}{W}\times V$$
where C_i_, C_t_, V and W are the initial metal concentration (mg L^−1^), concentration at time ‘t’, volume of solution (L) and adsorbent dosage (g), respectively. The zero point charge (pH_ZPC_) of samples was determined by plotting a graph between initial pH and ΔpH (pH_final_ − pH_initial_). The details of reusability studies and effect of coexisting cations and anions are discussed in Appendix A. All the experiments were run in three replicates, and average values are reported. Three error functions (Chi-square, sum of square of errors and residual root mean square error) were used to evaluate the validity of kinetic and isotherm models.

#### 3.3.2. Photocatalytic Experiments

The first experiment performed to determine the photocatalyst showing the highest activity. For this, 0.2 g L^−1^ of pristine MGF-B, BPGO, and HNFs were combined with 50 mL of malathion (2 mg L^−1^) at pH 3.0–9.0. In order to reach equilibrium between adsorption and desorption, the mixture was agitated for 30 min. A 16 W visible light emitting diode was then turned on, and the photocatalytic reaction began. The suspension parts from the reaction vessel were removed after two hours and centrifuged. The residual malathion concentration was studied by a UV-VIS spectrophotometer at the wavelengths of 250 nm. Then, using best photocatalyst, next photocatalytic experiments were performed. An aliquot of 5 mL was taken for kinetic analysis at regular intervals ranging from 2 min to 12 h. Using the Langmuir Hinshelwood model, the apparent rate constant, k, was calculated [47]:ln(C_o_/C_t_) = kt 
where C_o_ and C_t_ depicts organic pollutant concentration at initial time and after a definite period of light illumination, t stands for illumination time. The effect of the HNFs dose was investigated by changing the photocatalyst dose from 0.2 to 0.8 g L^−1^. The details of quenching and reusability experiments are discussed in the Appendix A. The photodegradation studies were carried out in triplicate, and an average of three results is reported to test the validity and repeatability of the analytical data.

### 3.4. Statistical Analysis Using Box–Behnken Methodology

Using Stat-Ease Design-Expert (Version 13) software, a Box–Behnken experimental design was used to determine the effect of independent variables (pH, dose, contact time, and temperature) and their simultaneous interactions on the response function (removal efficiency of Pb(II) ions and degradation efficiency of malathion) with three central points. The various experimental levels of independent factors are displayed together with their codes in Table 8. The following quadratic polynomial equation was employed for analysing and predicting the response:Y = β_o_ + Σ β_i_X_i_ + Σ β_ii_X_i_^2^ + ΣΣ β_ij_X_i_X_j_
where Y and X_m_ (m = i, j) stand for response and coded independent factors, respectively. β_0_ is zero-order constant, whereas, β_i_, β_ii_, and β_ij_ represents the linear, quadratic, and interaction coefficients of input independent variables, respectively. Selected design using independent variables “M” and central points “C_0_” proposed the N = 2M(M − 1) + C_0_ experiments to performed trial runs, where N represents the frequency of samples. Based on this, a total of 27 runs for the adsorption experiment and 15 runs for the photocatalytic experiment were planned for three coded levels (1, 0, +1) of chosen components.

Using the coefficient of determination (R^2^), adjusted coefficient of determination (Adj. R^2^), normal distribution of the residuals, and by plotting the actual values with predicted values, the suggested model’s quality and goodness were assessed. The probability critical level (*p*-value) of 0.05 was used in the analysis of variance (ANOVA), which was used to determine whether the parameters of the proposed model were statistically significant.

## 4. Conclusions

Hierarchical nanoflowers (HNFs) was successfully fabricated via ultra-sonication approach. It combined features of MgFe_2_O_4_-bentonitenanocomposite with boronand phosphorus doped GO. The synthesized HNFs displayed excellent adsorptive and photodegradation performance as compared to pristine components towards Pb(II) and malathion removal. This enhancement in adsorption and photodegradation was mainly attributed to reduced aggregation, specific surface area and remarkable charge transfer ability. Parametric optimization by Box–Behnken design was successfully achieved for adsorption and photocatalytic performance of HNFs for the removal of Pb(II) ions and degradation of malathion. The 96% removal of Pb(II) and 98% degradation of malathion was achieved using HNFs as adsorbent and photocatalyst, respectively. Hence, synthesized HNFs can serve as a promising candidate for wastewater remediation.

## Figures and Tables

**Figure 1 ijms-23-09678-f001:**
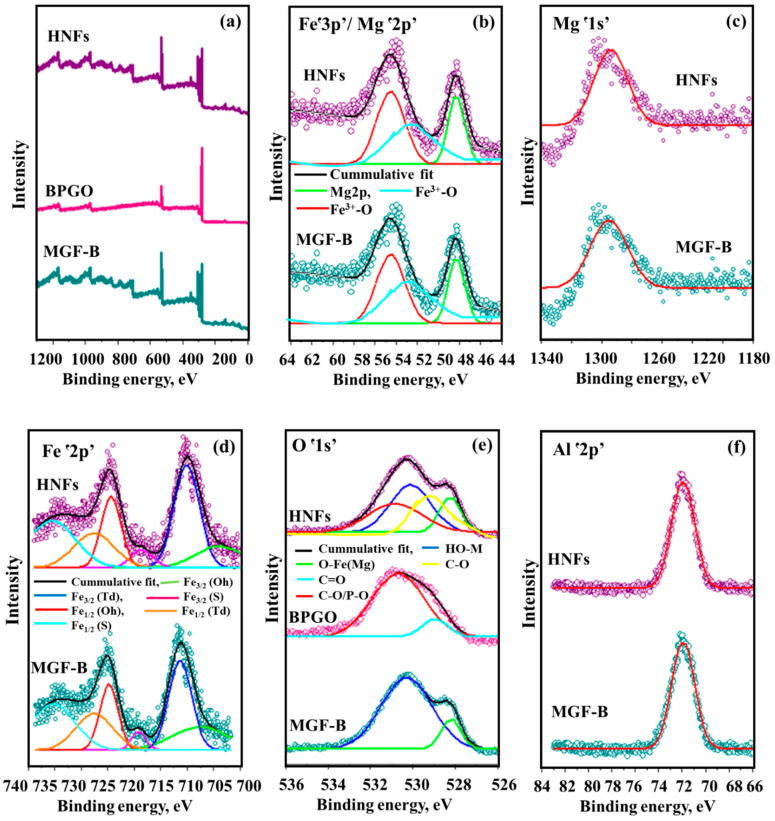
XPS spectra (**a**) Full scan spectrum and XPS high resolution spectra of (**b**) Mgՙ2p’/Feՙ3p’, (**c**) Mgՙ1s’, (**d**) Feՙ2p’, (**e**) Oՙ1s’ and (**f**) Alՙ2p’ of MGF-B, BPGO and HNFs.

**Figure 2 ijms-23-09678-f002:**
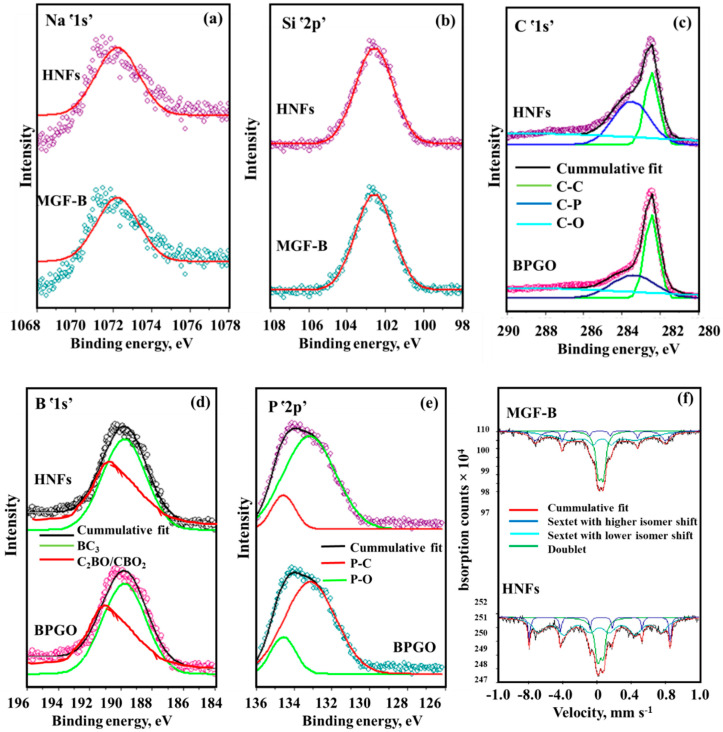
XPS high resolution spectra of (**a**) Naՙ1s’, (**b**) Siՙ2p’, (**c**) Cՙ1s’, (**d**) Bՙ1s’, (**e**) Pՙ2p’ and (**f**) Mössbauer spectra of MGF–B and HNFs.

**Figure 3 ijms-23-09678-f003:**
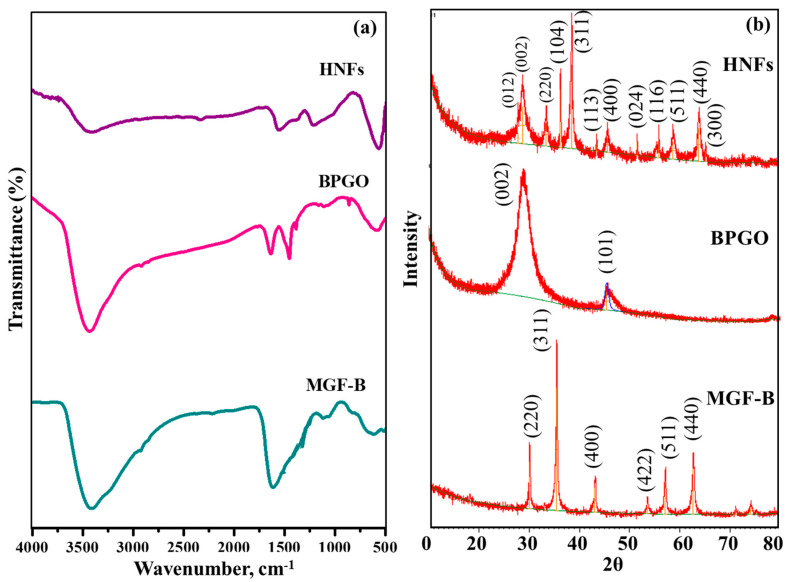
(**a**) FT–IR spectra and (**b**) XRD diffractogram of MGF–B, BPGO and HNFs.

**Figure 4 ijms-23-09678-f004:**
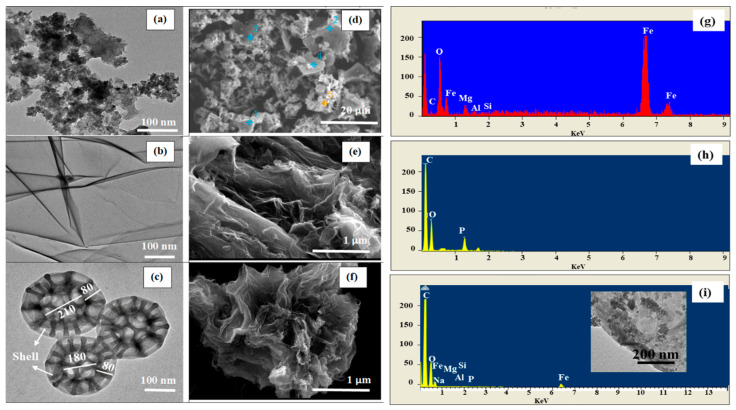
TEM images (**a**–**c**), SEM images (**d**–**f**), their EDS pattern (**g**–**i**) of MGF-B, BPGO and HNFs (Blue and yellow symbols in Figure 4d represents the points used for surface analysis using point and shoot method).

**Figure 5 ijms-23-09678-f005:**
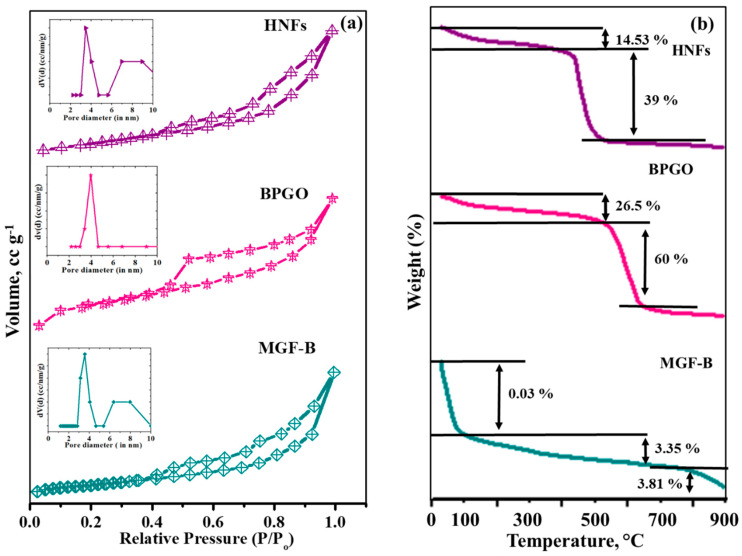
(**a**) BET isotherms, pore size distribution (inset) and (**b**) TGA plots of MGF–B, BPGO and HNFs.

**Figure 6 ijms-23-09678-f006:**
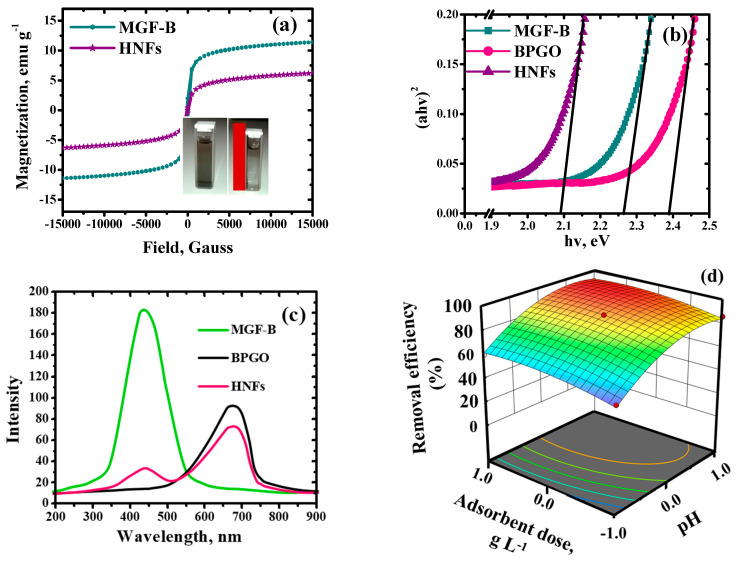
(**a**) VSM, (**b**) DRS plot, (**c**) photoluminescence spectra of MGF–B, BPGO and HNFs, Removal efficiency of Pb(II) as a function of (**d**) pH and adsorbent dose, (**e**) pH and contact time and (**f**) pH and temperature.

**Figure 7 ijms-23-09678-f007:**
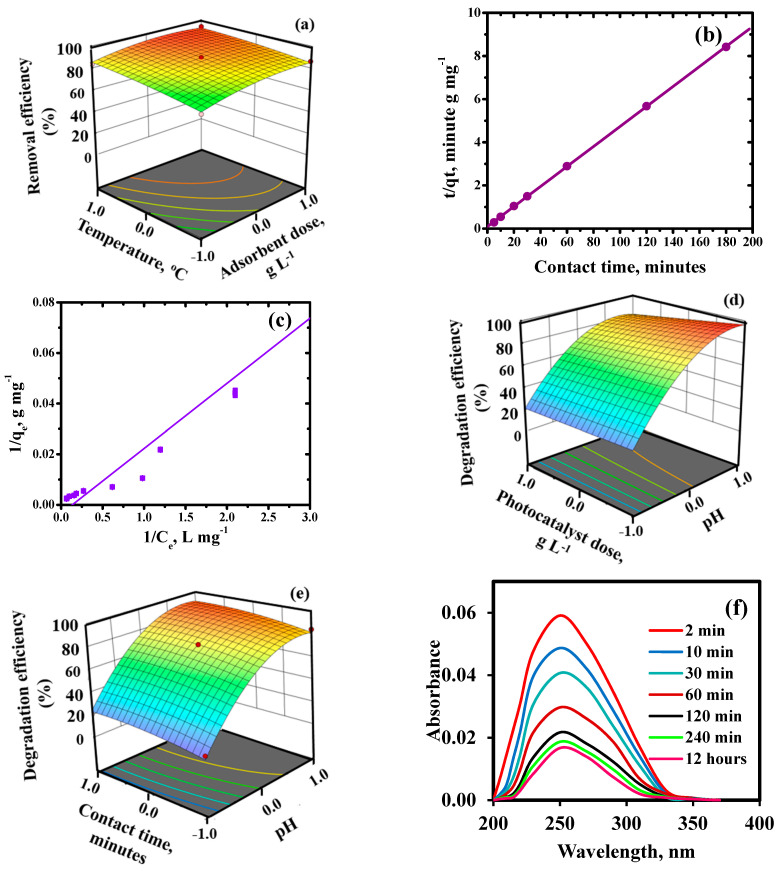
Removal efficiency of Pb(II) as a function of adsorbent dose and temperature (**a**), Pseudo–second order kinetic model (**b**), Langmuir adsorption isotherm (**c**), Degradation efficiency of malathion as a function of (**d**) pH and adsorbent dose, (**e**) pH and contact time and (**f**) temporal plots of malathion degradation. In b & c, straight lines represents fitting of model, where C_e_ stands for equilibrium concentration, q_e_ and q_t_ represents uptake capacity at equilibrium time “t” and at time “t”, respectively.

**Figure 8 ijms-23-09678-f008:**
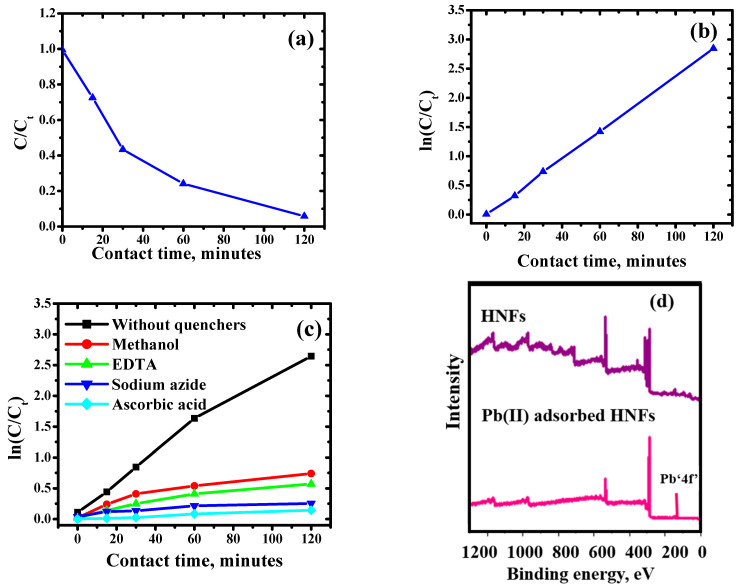
(**a**) The influence of contact time (**b**) kinetic plot (Experimental conditions: optimum pH, malathion concentration—2.0 mg L^−1^, HNFs dose—0.2 g L^−1^), (**c**) Kinetic studies of inhibition of degradation of malathion, (**d**) Full survey spectrum of HNFs and Pb(II)-adsorbed HNFs, (**e**) XPS high resolution spectra of Pbՙ4f’ and (**f**) FT–IR spectra of HNFs and Pb(II)-adsorbed HNFs. In (**a**,**b**) straight lines represents fitting of model, where C and C_t_ represents concentration of malathion at initial time “t” and at time “t”, respectively.

**Figure 9 ijms-23-09678-f009:**
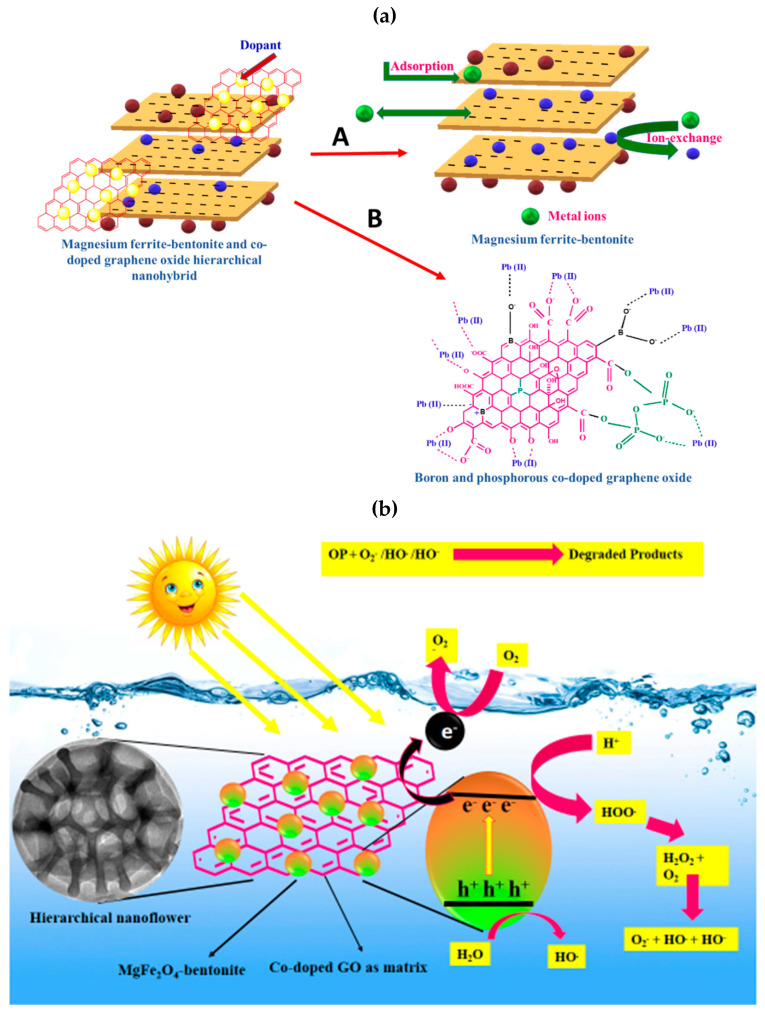
(**a**) Adsorption mechanism of Pb(II) ions by MGF-B (Path-A) and BPGO (Path-B) and (**b**) Photodegradation mechanism of malathion.

**Table 1 ijms-23-09678-t001:** Information regarding the sample codes.

Sample Code	Sample Description
MGF-B	Magnesium ferrite-bentonite nanocomposite
BPGO	Boron and phosphorous co-doped graphene oxide
HNFs	Hierarchical nanoflowers like morphology of nanocomposite containing magnesium ferrite-bentonite and boron and phosphorous co-doped graphene oxide

**Table 2 ijms-23-09678-t002:** Surface, magnetic, XRD parameters, and elemental composition by XPS and EDS analysis.

Properties		MGF-B	BPGO	HNFs
Elemental composition by XPS	C (at%)	-	84.48	59.22
	O (at%)	59.19	14.41	20.68
	B (at%)	-	0.27	0.19
	P (at%)	-	0.84	0.74
	Mg (at%)	24.26	-	12.70
	Fe (at%)	15.36	-	4.94
	Na (at%)	0.32	-	0.46
	Al (at%)	0.74	-	0.88
	Si (at%)	0.13	-	0.19
Surface	Pore Volume (cm^3^g^−1^)	0.07	0.24	0.20
	Surface Area (m^2^g^−1^)	87.1	162.81	157.56
	Pore Diameter (nm)	3.11	3.89	3.81
XRD	Lattice constant (Å)	0.83	0.70	0.81
	d-spacing (nm)	0.25	0.34	0.24
	Crystallite size (nm)	4.0	-	8.6
Elemental composition by EDS	O	48.54 (18.25)	16.29 (28.42)	21.93 (25.68)
	Mg	3.58 (2.70)	-	0.91 (1.40)
	Na	1.18 (0.15)	-	0.36 (0.12)
	Al	2.33 (0.15)	-	0.03 (0.07)
	Si	0.26 (0.28)	-	0.03 (0.16)
	Fe	44.11 (78.47)	-	1.25 (4.54)
	C	-	83.43 (71.41)	75.45 (67.83)
	P	-	0.28 (0.17)	0.04 (0.20)
Magnetic	Magnetization (emu g^−1^)	11.4	-	6.26
	Retentivity (emu g^−1^)	0.4	-	0.43
	Coercivity (Gauss)	138.0	-	70.65

Values in parentheses are weight percentages.

**Table 3 ijms-23-09678-t003:** Mössbauer parameters of magnesium ferrite-bentonite (MGF-B) and Hierarchical nanoflowers (HNFs).

	Sub Spectrum and Site	Isomer Shift (in mm s^−1^)	Quadruple Splitting (in mm s^−1^)	Hyperfine Field (in T)	Relative Area (in %)
MGF-B	Sextet (Td)	0.3 ± 0.02	−0.0 ± 0.04	45.3 ± 0.47	34.8
	Sextet [Oh]	0.3 ± 0.01	0.1 ± 0.02	48.5 ± 0.18	12.2
	Doublet	0.3 ± 0.00	0.6 ± 0.00	-	52.9
HNFs	Sextet (Td)	0.3 ± 0.02	0.1 ± 0.05	44.9 ± 0.29	62.5
	Sextet [Oh]	0.4 ± 0.00	−0.2 ± 0.01	51.4 ± 0.05	13.4
	Doublet	0.4 ± 0.02	0.7 ± 0.02	-	24.2

**Table 4 ijms-23-09678-t004:** The randomized design matrix for Box–Behnken methodology and observed responses for Pb(II) adsorption using HNFs.

Run	pH	Adsorbent Dose (g L^−1^)	Contact Time (min)	Temperature (°C)	Adsorption Efficiency
1	−1	0	1	0	60.4
2	0	−1	−1	0	80.3
3	−1	1	0	0	60.1
4	0	1	1	0	90.5
5	1	0	0	1	96.4
6	0	0	1	1	94.9
7	1	0	−1	0	92.4
8	0	0	0	0	90.2
9	1	0	1	0	92.4
10	1	1	0	0	92.7
11	−1	0	0	1	64.6
12	0	0	1	−1	82.7
13	0	0	0	0	90.2
14	0	−1	0	1	84.4
15	0	0	−1	1	94.1
16	0	−1	1	0	80.6
17	0	1	−1	0	90.7
18	−1	−1	0	0	54.6
19	−1	0	0	-1	56.4
20	0	0	−1	−1	82.1
21	0	0	0	0	90.2
22	−1	0	−1	0	60.3
23	0	−1	0	−1	72.5
24	0	1	0	1	94.5
25	1	−1	0	0	86.4
26	0	1	0	−1	86.5
27	1	0	0	−1	88.1

**Table 5 ijms-23-09678-t005:** Analyses of variance analysis of Pb(II) removal.

Source	Sum of Squares	Degree of Freedom	Mean Square	F-Value	*p*-Value	Status
Model	4650.23	14	332.16	129.71	<0.0001	Significant
A:pH	3072.00	1	3072.00	1199.64	<0.0001	
B: Adsorbent dose	275.52	1	275.52	107.59	<0.0001	
C: Contact time	0.0208	1	0.0208	112.36	<0.0001	
D: Temperature	300.00	1	300.00	117.15	<0.0001	
A.B	243.23	1	243.23	20.12	0.0021	
A.C	123.12	1	123.12	30.45	0.0034	
A.D	110.23	1	110.23	42.56	0.0043	
B.C	0.0625	1	0.0625	0.0244	0.8785	
B.D	4.00	1	4.00	10.56	0.0232	
C.D	0.0024	1	0.0024	0.0120	1.0000	
A^2^	886.95	1	886.95	346.36	0.0523	
B^2^	94.45	1	94.45	36.88	0.8412	
C^2^	3.70	1	3.70	1.45	0.02523	
D^2^	8.61	1	8.61	3.36	0.0916	
Residual	30.73	12	2.56	-	-	
Lack of Fit	30.73	10	3.07	4.52	0.86	Not significant
Pure Error	0.0024	2	0.0049	-	-	
Cor Total	4680.96	26	-	-	-	

R^2^ = 0.99, R^2^_adjusted_ = 0.98. Adequate Precision = 37.50.

**Table 6 ijms-23-09678-t006:** A randomized design matrix for Box–Behnken methodology and observed responses for malathion degradation using HNFs.

Run	pH	Photocatalyst Dose (g L^−1^)	Contact Time (min)	Degradation Efficiency
1	0	−1	−1	70.23
2	−1	1	0	20.24
3	−1	0	1	21.36
4	0	0	0	77.26
5	1	1	0	81.556
6	0	1	1	79.24
7	0	−1	1	84.29
8	−1	−1	0	24.21
9	1	−1	0	98.45
10	0	1	−1	63.56
11	1	0	−1	84.24
12	0	0	0	77.26
13	−1	0	−1	18.21
14	0	0	0	77.26
15	1	0	1	88.88

**Table 7 ijms-23-09678-t007:** An analyses of variance analysis of malathion degradation.

Source	Sum of Squares	df	Mean Square	F-Value	*p*-Value	Status
Model	11014.18	9	1223.80	79.73	<0.0001	Significant
A: pH	8978.00	1	8978.00	54.89	<0.0001	
B: Photocatalyst dose	136.12	1	136.12	8.87	0.0309	
C: Contact time	171.12	1	171.12	11.15	0.0206	
A.B	42.25	1	42.25	2.75	0.0158	
A.C	0.2500	1	0.2500	0.0163	0.0403	
B.C	1.0000	1	1.0000	0.0651	0.8087	
A^2^	1667.31	1	1667.31	108.62	0.0101	
B^2^	0.0030	1	0.0030	0.0010	1.0000	
C^2^	33.23	1	33.23	2.16	0.2012	
Residual	76.75	5	15.35	-	-	
Lack of Fit	76.75	3	25.58	5.36	0.75	Not significant
Pure Error	0.0030	2	0.0030	-	-	
Cor Total	11090.93	14	-	-	-	

R^2^ = 0.99, R^2^_adjusted_ = 0.98, Adequate Precision = 25.55.

**Table 8 ijms-23-09678-t008:** Experimental levels of independent factors and their codes in a Box–Behnken design.

Codes	Variables	Low Level (−1)	Central Point (0)	High Level (+1)
For adsorption experiments				
A	pH	2	6	10
B	Adsorbent dose (g L^−1^)	0.2	0.4	0.8
C	Contact time (min)	30	60	120
D	Temperature (°C)	10	25	50
For photocatalytic experiments				
A	pH	3	5	9
B	Photocatalyst dose (g L^−1^)	0.2	0.4	0.8
C	Contact time (min)	30	60	120

## Data Availability

Not applicable.

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
