# Peer review of "Hierarchical Nanoflowers of MgFe2O4, Bentonite and B-,P- Co-Doped Graphene Oxide as Adsorbent and Photocatalyst: Optimization of Parameters by Box–Behnken Methodology"

_ijms, 2022, doi:10.3390/ijms23179678_

Round 1

Reviewer 1 Report

The paper entitled: ”Hierarchical nanoflowers of MgFe2O4, bentonite and B-,P- co3 doped graphene oxide as adsorbent and photocatalyst: Optimization of parameters by Box-Behnken methodology, by Manpreet Kaur Ubhi, Manpreet Kaur, Dhanwinder Singh, Mohammed Javed Aderbal C. Oliveira, Vijayendra Kumar Garg and Virender K. Sharma, is very well conceptualized and clearly written. Some minor English errors must be corrected. I recommend the article acceptance.

Section 2.1: a table including information about samples codes would be more than welcome here. It is not accessible for the reader to search the information in the presented form of the paper.

Author Response

Thanks for the suggestions. Table 1 including information about the sample codes is incorporated in section 2.1.

Reviewer 2 Report

Review of the ijms-1867449 for the Authors: This article deals with the topic of hierarchical nanoformations for photocatalysis. The showed samples are interesting, both from the synthetic and morphological aspect as well as considering their applicability as adsorbents for Pb and photocatalysts for malathion degradation. Title and Abstract – Ok. Introduction – Offers a comprehensive intro for all the parts of the paper, why nanostructures, why graphene oxide, ferrites, why Pb and malathion, and why use BBD for optimization. Experimental and Results– Successfully detailed, really extensive and detailed, and with all of them you proved your starting hypothesise and rounded up to a conclusion. Conclusions – Summarizes your main findings nicely. The article really covered all the aspects, including the quenching and reusability studies, and offers a comprehensive scientific contribution to the field. Literature – Sufficient. The research is really quite well envisioned and detailed, and the article successfully reflects that. There are some technical upgrades necessary through the manuscript, but I guess proofreading can sort this out. The manuscript is well ordered and reads well. In my opinion the authors reasonably covered every aspect of the research in details. From the journal scope the topic itself is interesting so my recommendation is minor revision.

Author Response

Thanks for the positive outlook.
